# Inverse Dynamics Pretraining Learns Good Representations for Multitask Imitation

**David Brandfonbrener***
New York University

**Ofir Nachum**
Google

**Joan Bruna**
New York University

## Abstract

In recent years, domains such as natural language processing and image recognition have popularized the paradigm of using large datasets to pretrain representations that can be effectively transferred to downstream tasks. In this work we evaluate how such a paradigm should be done in imitation learning, where both pretraining and finetuning data are trajectories collected by experts interacting with an unknown environment. Namely, we consider a setting where the pretraining corpus consists of multitask demonstrations and the task for each demonstration is set by an unobserved latent context variable. The goal is to use the pretraining corpus to learn a low dimensional representation of the high dimensional (e.g., visual) observation space which can be transferred to a novel context for finetuning on a limited dataset of demonstrations. Among a variety of possible pretraining objectives, we argue that inverse dynamics modeling – i.e., predicting an action given the observations appearing before and after it in the demonstration – is well-suited to this setting. We provide empirical evidence of this claim through evaluations on a variety of simulated visuomotor manipulation problems. While previous work has attempted various theoretical explanations regarding the benefit of inverse dynamics modeling, we find that these arguments are insufficient to explain the empirical advantages often observed in our settings, and so we derive a novel analysis using a simple but general environment model.

## 1 Introduction

Pipelines in image recognition and natural language processing commonly use large datasets to pretrain representations that are then transferred to downstream tasks where data is limited [Devlin et al., 2018, Chen et al., 2020, Radford et al., 2021]. In this paper, we consider how this paradigm can be applied to imitation learning [Pomerleau, 1991, Ho and Ermon, 2016, Kostrikov et al., 2019]. In contrast to supervised learning domains where datasets consist of input-output pairs, imitation learning datasets consist of *trajectories* with both the input-output mapping to be learned (namely, observation-action pairs) as well as information about the dynamics of the environment. Given this additional structure, it is worthwhile to study pretraining approaches that can incorporate this structure to improve beyond methods from traditional supervised learning domains.

To formalize the precise notion of transfer between pretraining and finetuning phases, we consider a multitask imitation setting where the environment (i.e., the transition dynamics) is fixed and data is comprised of trajectories of *task experts* acting in this environment. A task is defined by a latent context variable that is observed by an expert demonstrator, but is not contained in the dataset, as shown in Figure 1. During pretraining, we have access to a large number of trajectories from various tasks, while during finetuning we have access to a small number of trajectories from a single task. The goal is thus to use the pretraining dataset to learn representations that contain information about the environment that facilitates efficient learning of the finetuning task.

---

*\*david.brandfonbrener@nyu.edu

37th Conference on Neural Information Processing Systems (NeurIPS 2023).

A number of existing works have proposed objectives for representation learning that are applicable in this setting [Schwarzer et al., 2021, Stooke et al., 2021, Yang and Nachum, 2021, Yang et al., 2023], and we consider a variety of algorithms and modes of analysis to determine which approach is the most promising. Algorithmically, we consider four generic classes of objectives for pretraining: inverse dynamics, behavior cloning, forward dynamics, and static observation modeling (Figure 1). We conduct two types of analysis. First, we conduct an extensive empirical evaluation and introspection of the candidate algorithms along with several strong baselines. Second, we present a simple but general theoretical model of the multitask representation learning problem and analyze the relative merits of the candidate algorithms under this model.

Our main results from these analyses are summarized as follows:

1. Across a broad array of experiments from visual observations in six environments, out of all approaches considered, inverse dynamics is the only one that consistently outperforms the baseline of training a model from scratch. The performance of inverse dynamics even matches that of finetuning from ground truth low-dimensional states on in-distribution contexts. Moreover, we find that inverse dynamics scales the best with pretraining dataset size and most effectively maintains relevant information about the observation in its learned representation.

2. In our simplified model of representation learning, we show that inverse dynamics pretraining efficiently recovers the ideal representation while behavior cloning can suffer from confounding and forward dynamics can suffer from poor sample efficiency. These results provide intuition for the empirical results and motivate why inverse dynamics pretraining is so performant and robust.

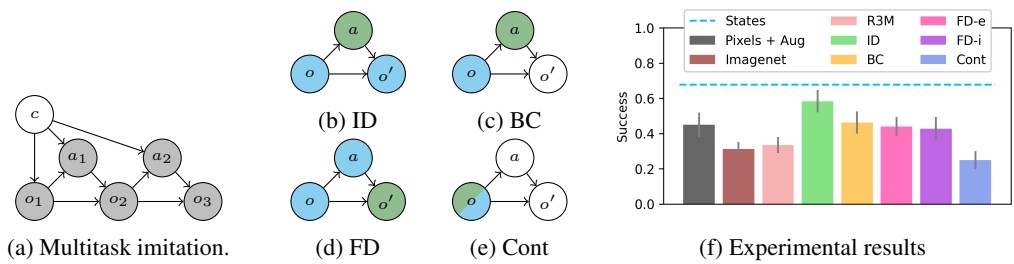

(a) Multitask imitation.    (b) ID    (c) BC    (d) FD    (e) Cont    (f) Experimental results

Figure 1: **(a)** A graphical models of the setting. Shaded nodes indicate observed variables. The expert behavior (i.e., $o_i \rightarrow a_i$) is determined by an unobserved context variable $c$ while the transition dynamics (i.e., $(o_i, a_i) \rightarrow o_{i+1}$) are determined by the environment dynamics. **(b)-(e)** illustrate the candidate algorithms. We use blue to indicate inputs to the algorithm and green to indicate prediction targets. ID = inverse dynamics, BC = behavior cloning, FD = forward dynamics, Cont = contrastive learning. **(f)** Shows success of policies finetuned on top of various representations averaged across all datasets in our suite for default dataset sizes. Inverse dynamics (shown in green) is the only method to substantially outperform the baseline of training from scratch (shown in black). Further details about the experimental protocol and results are in Sections 4 and 5.

## 2   Related work

As explained above, pretraining a representation has become a dominant paradigm in computer vision and natural language processing [Devlin et al., 2018, Chen et al., 2020, Radford et al., 2021]. Determining how to best leverage similar pretraining techniques in decision making problems is an important step towards extending the success of supervised learning into more temporally extended problems like those in robotics [Yang et al., 2023].

Prior work proposes several possible pretraining objectives for learning features for decision-making (and illustrated in Figure 1). First, inverse dynamics modeling has been proposed in several settings, although never as a representation learning algorithm for multitask imitation. Most directly related to our work is Efroni et al. [2021], Lamb et al. [2022] which use multi-step inverse dynamics for feature extraction for exploration in reinforcement learning (RL) in the presence of exogenous noise. Later work from Islam et al. [2022] extended this approach to offline RL. Less closely related are Pathak et al. [2017] which uses inverse dynamics in the context of exploration and Baker et al. [2022], Venuto et al. [2022] which use an inverse dynamics model to label video data with actions for imitation.

Another, perhaps simpler approach is to use behavior cloning as a pretraining algorithm. Arora et al. [2020] shows that this can be a well-motivated approach to pretraining a representation when the task variable is observed. Other work uses behavior cloning objectives to pretrain representations of temproally extended actions [Ajay et al., 2020] or priors for offline RL [Zang et al., 2022].

A third approach is to model the forward dynamics of the system as a pretraining objective. Most directly related to our work, Nachum and Yang [2021] show that this is a well-motivated technique for imitation learning and provide empirical evidence on single task atari games, but do not compare to inverse dynamics. This technique has also been explored in empirical work for online and offline RL [Schwarzer et al., 2021, Laskin et al., 2020, Aytar et al., 2018, Lee et al., 2022b, Wu et al., 2023].

Finally, a method which we will refer to as static observation modeling does not leverage information about dynamics and rather directly uses self-supervised methods from computer vision [Pari et al., 2021, Chen et al., 2020, Grill et al., 2020]. This approach does not take advantage of any additional structure in an imitation learning setting, but has nevertheless worked well in some settings.

Several empirical studies of representation learning for decision-making already exist. Most closely related to this work, [Chen et al., 2022] conducts an empirical evaluation of representations for imitation and finds that none of them consistently outperform training directly from pixels. However, this prior work (a) considers much larger finetuning datasets which can dramatically reduce the benefits of pretraining, and (b) considers different environments than we do, where the gap between pretraining and finetuning tasks is less controlled. Another line of work like Nair et al. [2022] attempts to pretrain general representations using large human-collected video datasets like Ego4d [Grauman et al., 2022]. In contrast, we focus on a more carefully controlled (albeit smaller scale) experimental settings where we can derive a more clear understanding of the relative merits of different pretraining objectives. Another empirical study from Stooke et al. [2021] considers representations in online reinforcement learning. Meanwhile, Yang and Nachum [2021] considers representations for imitation but does not consider image-based or multitask problems. Moreover, none of these works includes a theoretical understanding for the findings presented therein.

A further discussion of pretraining in the context of imitation can be found in Appendix A.

# 3 Problem setup

Here we present the formal setup for our problem setting of reward free pretraining from multitask expert data . We formalize this as a contextual MDP with rich (i.e., visual) observations where the latent context determines the initial state and reward functions.

**Environment.** We model the environment as a contextual MDP with context-independent dynamics:

$$c \sim P_c, \qquad o_0 \sim \rho_c, \qquad r_i = r_c(s_i, a_i), \qquad o_{i+1} \sim T(o_i, a_i). \qquad (1)$$

Importantly, we consider the context variable $c$ and rewards $r_c$ to be *latent*, i.e., they are not available during training, and only used to evaluate a learned policy. At a high level, this captures the setting where the task (defined by the context variable) may change, but the dynamics of the world do not. For example, the context variable could be a continuous variable like a goal position that the expert is navigating towards or a discrete variable representing a behavior like locking a door.

**Data generation.** Data is generated by executing policies $\pi$ that map observations to actions in the environment. We consider two different datasets for any given problem. First there is a large multi-context pretraining dataset that will be used for representation learning, specifically to learn an observation encoder. Second, there is a small single-context finetuning dataset for policy learning on top of the pretrained representation. The multi-context pretraining data is generated as follows:

$$D_{pre} = \{\tau_i\}_{i=1}^{N_{pre}}: \quad c \sim P_c, \quad \tau = (o_0, a_0, o_1 \dots) \sim P^{\pi_c}, \quad \pi_c \approx \pi_c^* = \arg\max_\pi J_{r_c}(\pi), \quad (2)$$

where $J_{r_c}(\pi)$ denotes the expected return of $\pi$ when the reward is $r_c$. Note that the demonstration policy has access to the latent context $c$, but this latent context is not observed in the data.

Then the single-context finetuning data is generated for context $c_{fine}$ as follows:

$$D_{fine} = \{\tau_i\}_{i=1}^{N_{fine}}: \tau = (o_0, a_0, o_1 \dots) \sim P^{\pi_{c_{fine}}}. \qquad (3)$$

**Pretraining.** The goal of the paper is to analyze different methods for pretraining feature extractors. Training of the encoders $\phi$ to minimize a loss $\ell$ proceeds as follows:

$$\hat{\phi} : \mathcal{O} \to \mathbb{R}^d = \arg\min_{\phi} \mathbb{E}_{D_{pre}} [\ell(\phi, \tau_i)]. \tag{4}$$

A full description of the losses $\ell$ used by different algorithms will come in Section 4.2. For simplicity (and in keeping with prior work [Nachum and Yang, 2021, Chen et al., 2022]) we will consider $\ell$ to only be a function of transitions $(o_i^j, a_i^j, o_i^{j'})$ rather than full trajectories to leverage the Markovian structure. We also run some ablations of including multistep information in Appendix B and find little difference.

**Finetuning.** Features are evaluated by finetuning a small policy head on top of the frozen features:

$$\hat{\pi}_{\hat{\phi}} : \mathbb{R}^d \to \mathcal{A} = \arg\min_{\pi} \mathbb{E}_{D_{fine}} [\ell(\pi, a_i^j, \hat{\phi}(o_i^j))]. \tag{5}$$

In all of our experiments, $\ell$ is the mean squared error loss for behavior cloning. We elect to use frozen features to allow for simple and clear evaluation of the representations. This is in keeping with prior work on representations for imitation [Nachum and Yang, 2021, Chen et al., 2020, Nair et al., 2022] as well as computer vision [Chen et al., 2020].

**Evaluation.** Finally, we evaluate the finetuned policy by performing rollouts in the finetuning environment with context $c_{fine}$ to estimate $J_{r_{c_{fine}}}(\hat{\pi}_{\hat{\phi}})$. In our tasks we usually consider $r_{c_{fine}}$ to be a binary indicator of successful completion of the finetuning task.

## 4 Experimental setup

### 4.1 Environments and Datasets

We design a suite of tasks and datasets to probe the capabilities of various representation learners for downstream imitation. We focus on robotic manipulation from vision as this is an important sequential decision making task that depends on learning task-relevant visual representations where pretraining deep visual feature extractors is a popular approach. Our suite consists of six different pretraining datasets on varied tasks and of varied size. Each pretraining dataset has several associated finetuning datasets and simulation environments that allow for online evaluation of learned policies.

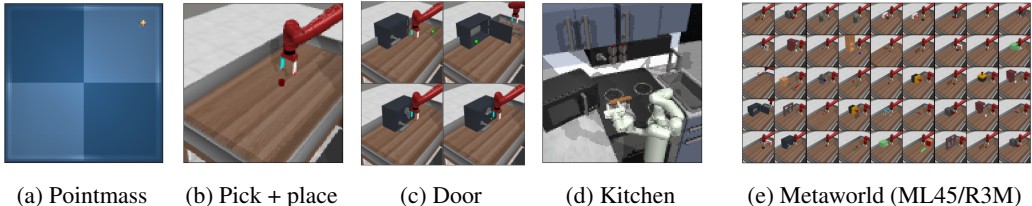

| (a) Pointmass | (b) Pick + place | (c) Door | (d) Kitchen | (e) Metaworld (ML45/R3M) |

Figure 2: Our six datasets: (a) Pointmass navigation with latent goals. (b) Pick and place with latent goals. (c) Multitask manipulation of a door. (d) Sequential kitchen manipulation. (e) Multitask manipulation of diverse objects, where we consider two different train-eval splits ML45 and R3M.

All tasks are performed from visual inputs, as shown in Figure 2. Each pair of pretraining-finetuning datasets requires a slightly different type of generalization as dictated by the different types of context variable. Specifically, in the pointmass and pick+place datasets the context variable is a latent goal position, while in the door and metaworld datasets the context variable is a discrete identity of a desired behavior, and in the kitchen datasets the context variable is a discrete ordered sequence of subtasks. The datasets are described in full detail in Appendix C.

### 4.2 Algorithms

We consider nine different representations across our suite of experiments. These representations include baseline and skyline/oracle performance as well as five representations that are pretrained on

Table 1: Description of the different datasets used in the experiments. Dataset sizes are measured in number of trajectories ($N_{traj}^{pre}$ for pretraining and $N_{traj}^{fine}$ for finetuning) and given as ranges with default values in bold. Trajectory lengths vary from 50 to 400 steps. These default sizes may vary in each experiment when indicated. Each datasets contains a certain number of latent contexts ($N_{context}^{pre}$ and $N_{context}^{fine}$). For each finetuning context, we sample datasets with $N_{seed}^{fine}$ different seeds.

| Environment | $N_{traj}^{pre}$ | $N_{traj}^{fine}$ | $N_{context}^{pre}$ | $N_{context}^{fine}$ | $N_{seed}^{fine}$ |
|---|---|---|---|---|---|
| Pointmass | (1e1, 1e2, **1e3**) | (1, **2**, 5, 10) | $N_{traj}^{pre}$ | 5 | 1 |
| Pick + place | (1e1, 1e2, **1e3**) | (2, **5**, 10, 20) | $N_{traj}^{pre}$ | 5 | 1 |
| Door | (1e1, 1e2, **1e3**) | (2, 5, **10**, 20) | 3 | 1 | 5 |
| Kitchen | (50, 150, **450**) | (2, 5, **10**, 15) | 21 | 3 | 5 |
| MW-ML45 | (1e2, 1e3, **1e4**) | (2, 5, **10**, 20) | 45 | 5 | 5 |
| MW-R3M | (1e2, 1e3, **1e4**) | (2, 5, **10**, 20) | 45 | 5 | 5 |

our own pretraining datasets described above. Each of the representations will be referred to by its bolded name after it is described.

All algorithms (except for the Imagenet and R3M baselines) share the exact same encoder architecture to control as best we can for variation in architecture between methods. Each method is pretrained for the same number of gradient steps. Additional training details can be found in Appendix C.

**Skyline/oracle.** As a skyline or oracle representation we directly use the low dimensional states (**States**) from the simulator. Depending on the task, this representation includes the position of the robot, position of the object to be manipulated, and/or position of the goal. A full description of the per environment state variables can be found in Appendix C.

**Baselines.** We consider three baseline representations that are not trained on our pretraining datasets. The first is to directly use the pixels with image augmentations (**Pixels + Aug**) to train an encoder and a policy from scratch on the finetuning data. It is essential to use the augmentations to ensure that this a strong baseline. The second is features of a ResNet18 pretrained on Imagenet (**Imagenet**). The last consists of the features of a ResNet18 that is specifically pretrained for robotic manipulation by Nair et al. [2022] on the Ego4d dataset (**R3M**).

**Inverse dynamics.** The primary representation learning objective that we consider is inverse dynamics (**ID**) which models the distribution $P(a|o, o')$ using an architecture that first encodes $o, o'$ with an encoder $\phi$ and then predicts $a$ with a small MLP $f$:

$$\phi_{ID}^* = \arg\min_\phi \min_f \mathbb{E}_{o,a,o'} [(a - f(\phi(o), \phi(o')))^2]. \tag{6}$$

**Behavior cloning.** A simpler alternative objective is to directly apply behavior cloning (**BC**) to the multitask actions in the pretraining dataset conditioned on the observations using MSE loss. The learner is parameterized as an encoder $\phi$ followed by a small MLP $\pi$:

$$\phi_{BC}^* = \arg\min_\phi \min_\pi \mathbb{E}_{o,a} [(a - \pi(\phi(o)))^2]. \tag{7}$$

**Forward dynamics.** We consider two representation learners that predict the forward dynamics of the system. The first is explicit forward dynamics (**FD-e**) which explicitly constructs a model of the forward dynamics in the space of observations by encoding the current observation and then attempting to reconstruct the next observation $o'$ using a decoder $d$:

$$\phi_{EFD}^* = \arg\min_\phi \min_d \mathbb{E}_{o,a,o'} [(o' - d(\phi(o), a))^2]. \tag{8}$$

The second objective is implicit forward dynamics (**FD-i**) which implicity constructs a model of the forward dynamics using contrastive learning. Explicitly, we consider a form of contrastive learning where an energy function is defined as the inner product of L2-normalized projected embeddings (given by projection MLPs $f_1, f_2$) which is then passed into an InfoNCE loss:

$$E(o, a, o') = \exp(f_1(\phi(o), a)^\top f_2(\phi(o'))), \tag{9}$$

$$\phi_{IFD}^* = \arg\min_\phi \min_{f_1, f_2} \mathbb{E}_{o,a,o'} [-\log(E(o, a, o')) + \log \mathbb{E}_{\bar{o}'}[E(o, a, \bar{o}')]]. \tag{10}$$

**Static observation modeling**    Finally, we consider a baseline that simply models $P(o)$. Rather than modeling this explicitly with reconstruction, we use a contrastive loss (**Cont**) where we use image augmentations to construct pairs of $o$ and $\bar{o}$ that do not rely on the dynamics of the environment at all. Again we use the InfoNCE loss, in what can be seen as a variant of SimCLR:

$$E(o, o_{aug}) = \exp(f(\phi(o))^\top \pi(\phi(o_{aug}))), \tag{11}$$

$$\phi^*_{Cont} = \arg\min_\phi \min_f \mathop{\mathbb{E}}_{o, o_{aug}} [-\log(E(o, o_{aug})) + \log \mathop{\mathbb{E}}_{\bar{o}_{aug}} [E(o, \bar{o}_{aug})]]. \tag{12}$$

## 5    Experiments

We want to determine which representation learning objective is best, but the precise answer will depend on the situation. To get a clearer understanding of this sometimes ambiguous performance we conduct a variety of controlled experiments on our diverse suite of datasets. We focus on the following questions to guide our empirical analysis:

1. How do factors of the datasets impact performance of algorithms?
2. How are the learned representations similar to and different from each other?

Note: we will focus on presenting aggregate statistics across all datasets in the main text, but full results can be found in Appendix B. Full details about the methodology are in Appendix C and code is at `https://github.com/davidbrandfonbrener/imitation_pretraining`.

### 5.1    Impact of dataset on representation learning performance

**Scaling with data size.**    The performance of each algorithm can be highly sensitive to both pretraining and finetuning sizes. Thus, instead of producing one simple summary statistic, we sweep over both the size of the finetuning data (for default pretraining size) and size of the pretraining data (for default finetuning size). The results of these sweeps are presented in Figure 3.

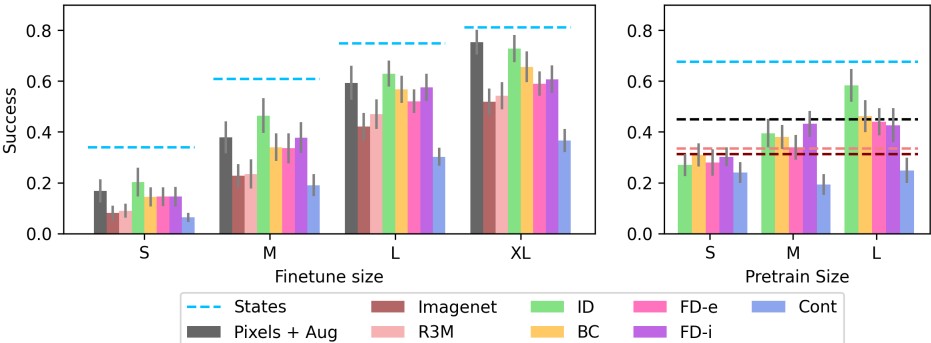

Figure 3: Average success rate after finetuning averaged across datasets, contexts, and seeds. Error bars show the standard error across contexts and seeds, averaged across datasets. The plots show sweeps across finetuning size with default pretraining size (left) or pretraining size with default finetuning size (right) measured in units according to Table 1. Methods that do not depend on pretraining size are shown as horizontal lines.

The sweeps both suggest that inverse dynamics outperforms the alternatives. First, on the finetuning size sweep, we see that the ID line is the only one that consistently outperforms training from scratch on Pixels + Aug. This gap is largest at small finetuning sizes, which are perhaps the most interesting case since that is when we expect pretraining to be useful. Second, the pretraining size sweep indicated that ID is scaling the most efficiently with pretraining size. Further results, including breakdowns across each dataset can be found in Appendix B.

**In distribution vs. out of distribution eval tasks.**    The way that our datasets are constructed, the door, kitchen, metaworld-ml45, and metaworld-r3m datasets only have a finite number of possible contexts that is much smaller than the number of pretraining trajectories. For our default datasets, we elected to construct a train-test split of contexts to ensure that the contexts used for finetuning are

not seen during pretraining. As a result, the default finetuning tasks can be in some sense "out of distribution", measuring extrapolation as opposed to in-distribution generalization. For example, in the door dataset, we pretrain on door opening, closing, and unlocking (with varied door position) and then finetune on door locking (again with varied position).

To test the impact of this gap between pretraining and finetuning, we created alternative pretraining datasets, where we include the test contexts (but *not* the test trajectories) into the pretraining data. For example, in the door domain we include door opening, closing, locking, *and* unlocking in the pretraining data and still finetune on only unlocking (but with heldout initial conditions). These datasets now require a much more limited notion of generalization from pretraining to finetuning.

Results are shown in Figure 4. We again see that ID is the strongest performer, but now the gap is even larger. ID matches the skyline performance of training from ground truth low-dimensional simulator states. BC also shows substantially stronger performance and outperfroms training from scratch Pixels + Aug. None of the other pretraining algorithms benefit much from the substantially easier type of generalization required on these datasets. This suggests that ID and BC are uniquely able to benefit in easier settings, suggesting that they are better representation learners. If an algorithm is not able to outperform training from scratch in this simplified setting, it is unlikely to be a good representation learner.

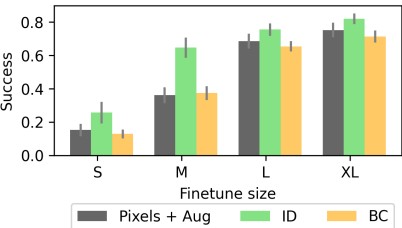

Figure 4: Average performance on the four discrete context environments when the finetuning contexts are included in the pretraining data. The finetuning data contains heldout initial conditions and trajectories not seen during pretraining.

**Fully latent vs. inferrable context variables.** Looking at our dataset suite, the datasets can be divided into two groups: those where the context variable is not inferrable at all from the initial state (pointmass, pick+place, and kitchen), and those where the effect of the context variable on the initial state makes it possible to infer the context given the initial state (door, metaworld-ml45, and metaworld-r3m). This split presents an interesting comparison in particular between ID and BC (the best performing algorithms from the prior experiment). Figure 5 shows results for these algorithms and the Pixels + Aug baseline on the datasets where the context is latent.

Figure 5: A comparison between ID and BC on the datasets where the context is not inferrable from the observation.

There is a large gap between ID and BC when the context is fully latent. In these cases, it is impossible to tell from the current state alone what the context is and thus what the optimal action should be. As we will show in our simplified model (Section 6), in these settings BC is *confounded* by the latent context (in the terminology of causal inference). As a result, BC can fail to learn useful features. In contrast, ID uses the information about the future state to deconfound the learning problem and still learns a good representation. Note that this gap largely disappears when the context is observable, see Appendix B for further details.

## 5.2 Predictive power of the representations

So far, we have focused on the success rate of the downstream finetuned policy as the main metric of comparison between algorithms. Now we will instead consider a series of experiments that assess the quality of the representations based on the ability to predict various quantities of interest from the representations. These experiments help to illustrate what information is retained in the representations and how efficiently that information can be accessed.

**Action prediction.** First, we consider the ability to predict the expert actions in the finetuning dataset. This is directly related to the success of the finetuned policy, but avoids the variance of

performing rollouts and allows us to compare train and validation errors to evaluate the representations. Low train loss means the representations are not aliasing observations that require different actions. Meanwhile the validation loss measures the simplicity of the function that maps representations to targets, i.e. how well it generalizes.

The results in Figure 6 show the train and validation loss during finetuning using the default pretraining and finetuning sizes from Table 1. Since losses vary across datasets, we normalize by the Pixels + Aug validation loss so as to be able to present averages across all datasets. We see that out of the learned representations, ID has both the lowest train and validation losses, almost matching the performance of Pixels + Aug on train and almost matching the performance of States on validation. In contrast, representations that attempt to predict forward dynamics have substantially higher train loss, indicating aliasing of states in terms of their optimal actions. Interestingly, the Imagenet pretrained features have very low train loss, indicating a lack of aliasing, but very high validation loss, indicating that the function that maps representations to actions does not generalize well.

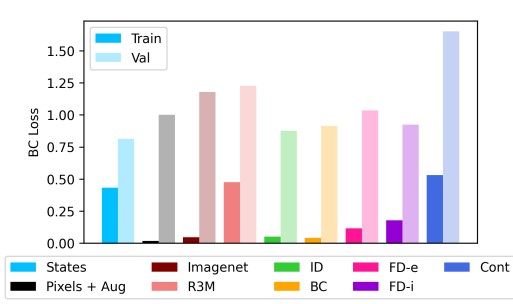

Figure 6: Average train and validation action-prediction loss during finetuning. All losses are normalized by the Pixels + Aug validation loss to maintain consistency across environments.

**State prediction.** Since we perform all of our experiments in simulated environments, we have access to the ground truth low dimensional states. So, we can measure the ability of each representation to predict the ground truth low dimensional state and thus measure how well the representation retains information about this ground truth state. Results are in Figure 7; here we measure the train and validation loss on the pretraining distribution so as to isolate the effect of the representation learning apart from the gap between pretraining and finetuning. Again we normalize the losses for each dataset.

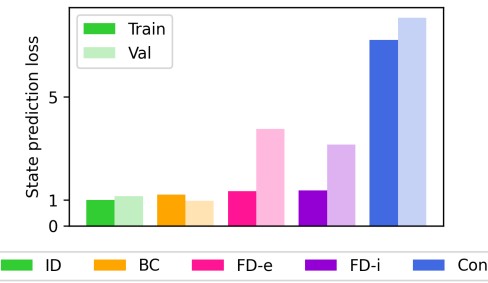

Figure 7: Average state prediction error on the pretraining distribution. Values are normalized by the ID train loss.

Again we see that ID and BC yield the best performance. This suggests that in these datasets, pretraining objectives that attempt to predict the optimal action do indeed facilitate recovery of the low-dimensional simulator state. In contrast, while the FD methods achieve approximately the same training error, they generalize much more poorly. This suggests that the FD objectives are not throwing away relevant information, but are keeping around too much extraneous information about the observations, thus making the representations susceptible to overfitting. Standard contrastive learning is substantially worse, even on train error, suggesting that it is throwing away important information. Extended results are in Appendix B.

# 6   Analysis

To add a more theoretical understanding of the empirical results, we will consider a simplified model of the data generating process based on linear dynamics in a latent space. We begin by presenting the model and then show that under this model we can explain three key experimental findings: (1) inverse dynamics is able to recover the low dimensional state, (2) forward dynamics can be less efficient in some cases, and (3) BC can be confounded by the latent context. We present a high level sketch here and more details along with discussion of related theoretical work are in Appendix D.

**Model.**   Some of the key interesting properties of problems like visual manipulation that we consider empirically are that (a) the observation is very high dimensional relative to the action, (b) the actual state of the world (or simulator) can be summarized in a much lower dimensional state variable, and (c) the dynamics are relatively simple if given the right representation. All of these motivating

properties can be captured in a simplified model that assumes linear dynamics occurring in a hidden low-dimensional state space, as presented below.

For simplicity, we will only consider one step of the dynamics represented by a tuple $(o, a, o', s, s', c)$ that is sampled iid from the joint distribution over those variables. Recall that we only observe $(o, a, o')$ and that $(s, s', c)$ are latent. Formally, let $\mathcal{O} = \mathbb{R}^d$, $\mathcal{S} = \mathbb{R}^\ell$, and $\mathcal{A} = \mathbb{R}^k$ with $d \gg \ell > k$. Let $\phi : \mathcal{O} \to \mathcal{S}$ be the ground truth encoder, which we assume is invertible by $\phi^{-1}$. Let $\epsilon \sim \mathcal{N}(0, \Sigma)$ in $\mathbb{R}^\ell$ and $A, B$ to be any matrices in $\mathbb{R}^{\ell \times \ell}$ and $\mathbb{R}^{\ell \times k}$. Then, assume that the dynamics are:

$$o' = \phi^{-1}(A\phi(o) + Ba + \epsilon). \tag{13}$$

Note that we make no assumption on the policy $\pi_c^*$ other than that it only depends on $o$ via $\phi(o)$. This model is similar to ones studied in the online control setting by Mhammedi et al. [2020], Dean and Recht [2021], but is different from models where inverse dynamics have been studied for online control with exogenous noise since the dynamics are entirely contained in the low dimensional state space [Efroni et al., 2021, Lamb et al., 2022].

**Inverse dynamics recovers the state.**   To get an intuition as to why inverse dynamics learning is feasible in this model, note that if $B^+$ is the pseudoinverse of $B$ that:

$$a = B^+\phi(o') - B^+A\phi(o) - B^+\epsilon. \tag{14}$$

Thus the inverse dynamics are a simple linear function of the embeddings $\phi(o), \phi(o')$. As a result, when we solve for $a$ with least squares regression, if the encoder $\phi$ is representable by our function class, we will be able to recover it up to linear transformation, provided the matrix $B$ is well-conditioned, so that the noise term $B^+\epsilon$ does not blow up.

**Forward dynamics can be less statistically efficient.**   Intuitively, the potential problem with learning forward dynamics is that it requires learning both an encoder *and* a decoder while inverse dynamics *only* requires learning the encoder. This is not necessarily a problem a priori, but we hypothesize that in practical problems of interest (like the ones in our experiments) the decoder (mapping from low dimensional state to high dimensional observation) may be more complicated than the encoder (mapping from observations to states).

To grasp why we might expect this, note that the set of possible observations is the manifold represented by the image of the decoder, i.e. $\text{Im}(\phi^{-1})$. As a simple example, consider a toy 2d example where the high dimensional observation is $(x, f(x)) \in \mathbb{R}^2$ and the low dimensional state is simply $x \in \mathbb{R}^1$, as depicted in Figure 8. Here the encoder $\phi$ is very simple since it just needs to recover $x$, while the decoder must learn $f(x)$. Of course this is a very toy example, but we find it illustrative of the idea that it is possible that the encoder is much simpler than the decoder in practice.

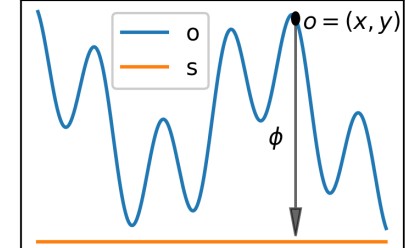

Figure 8: An example where the decoder is more complicated than the encoder.

**BC can be confounded by the latent context.**   As we alluded to in the experimental section, the latent context variable can confound BC. Now we will show an example in our model where this problem arises. In this case, even with a linear encoder, infinite data, and a fully expressive policy class, the Bayes optimal BC representation cannot be used to recover anything better than a random policy. This example is extreme, but shows the shortcomings of a confounded pretraining objective.

For simplicity, let $\ell = k$ and $\epsilon = 0$. Let $\mathcal{R}(\mathbb{R}^{k \times k})$ be the set of rotation matrices in $\mathbb{R}^k$. Let $\mathbb{S}^{k-1}$ be the unit sphere in $\mathbb{R}^k$, $U$ be the uniform distribution, and $\delta$ denote a Dirac delta. Now, assume:

$$c \sim U(\mathcal{R}(\mathbb{R}^{k \times k})), \quad o \sim U(\phi^{-1}(\mathbb{S}^{k-1})), \quad \pi_c^*(a|o) = \delta[a = c\phi(o)] \tag{15}$$

Note that $\phi(o)$ returns a unit vector in $\mathbb{R}^k$ and that a uniformly sampled rotation of a unit vector is a uniformly sampled unit vector. Thus, we can marginalize over $c$ to get:

$$P(a|o) = \int_c P(c)\pi_c^*(a|o) = \int_c P(c)\delta[a = c\phi(o)] = P_{U(\mathbb{S}^{k-1})}(a) = \eta_k, \tag{16}$$

for a constant $\eta_k$ equal to the reciprocal of the surface area of the unit sphere in $\mathbb{R}^k$.

Thus, the Bayes optimal BC policy does not depend on $o$ at all. As a result, the optimal representation learned by BC can just map every observation to zero. This representation is not capable of representing the optimal policy for any choice of $c$. However, switching to inverse dynamics pretraining where we condition on the outcome observation $o'$ breaks the confounding and allows us to learn the true representation even without observing $c$.

## 7   Discussion

We have seen that inverse dynamics pretraining provides an effective method for learning features from multitask demonstration data. We demonstrated this across a suite of datasets with visual observations and provided analysis in a simplified model to understand the strong empirical performance.

**Limitations.**   There are still a few limitations of our work that are worth pointing out explicitly. First, in this work we prioritized simulated domains with large numbers of predefined tasks and datasets with a single morphology to allow for a variety of experiments. However, it is possible that the results we observed in these tasks would differ when scaled to real world tasks with additional visual diversity and physical realism. We leave this extension to future work.

Second, while our theoretical analysis provides a clear rationale for the observed empirical results in a toy model, there is clearly room for better theory. Ideally, future work could present a more rigorous theory that goes beyond a toy model. However, we do think that the toy model captures some of the essential characteristics of the problem and recognize that any theory must make simplifying assumptions.

**Future directions.**   In addition to removing the limitations described above, there are many other interesting directions for future work to build on our results. One direction would be to extend these results to settings with suboptimal data. In this work we focus on an imitation learning setting where data is collected by expert policies across a variety of tasks. In future, it would be interesting to study how and if the properties of various representation learning algorithms change in the presence of suboptimal data.

It would also be interesting for future work to compare the relative merits of a broader array of pretraining techniques that go beyond representation learning. For example, methods that learn conditional generative models (e.g. goal-conditioning, language-conditioning, or reward-conditioning) provide a different paradigm for pretraining policies instead of the feature extractors that we consider in this work.

Finally, it would be interesting to consider developing new pretraining objectives for representation learning. This could be done by combining existing objectives or developing completely new ones.

### Acknowledgments

This work was completed as part of the Google Research Collabs program. We would like to thank Mahi Shafiullah, Mark Goldstein, Aahlad Puli, Siddhant Haldar, Ben Evans, Ulyana Piterbarg, and Lerrel Pinto for helpful discussions and feedback.

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

# A  Extended related work

In this paper we focus specifically on pretraining methods that learn representations of high dimensional observations from multitask demonstration data with latent contexts for the purpose of imitation. There are many closely related problems that are studied in other papers that we did not have space to address fully in the main text that we more fully describe here. These are all very interesting and complementary lines of work, but are beyond the scope of this paper.

Perhaps the largest closely related line of work focuses on learning reward-directed representations in the context of reinforcement learning. This is a different setting than ours and methods from there are not applicable in our setting where we do not have access to rewards. Moreover, most of these methods do not consider multitask settings [Zhang et al., 2020, Gelada et al., 2019, Fu et al., 2021, Ghosh et al., 2018, Eysenbach et al., 2022, Sodhani et al., 2021].

Another line of work seeks to learn representations of actions or sequences of actions rather than observations. This is a directly complementary line of work to the ideas presented in this paper [Ajay et al., 2020, Yang et al., 2021, Lynch et al., 2020, Whitney et al., 2019].

Another body of literature focuses on learning representations that can be transferred across domain and embodiment gaps and even trained directly from videos without access to actions at all [Oord et al., 2018, Aytar et al., 2018, Seo et al., 2022, Ma et al., 2022, Zakka et al., 2022, Ghosh et al., 2023]. In this paper, we focus on the simpler task of pretraining a representation within one MDP with consistent dynamics and access to demonstration actions, but with varied tasks. This choice allows us to make more clear comparisons between algorithms and rigorous claims about when representations will be effective, but also limits the generality of the representations that are learned.

There are a variety of new methods that rely on transformer architectures to construct interesting new pretraining objectives [Yang and Nachum, 2021, Lee et al., 2022a, Reed et al., 2022, Seo et al., 2023, Wu et al., 2023]. In this paper we focus on simple methods that can all use the same simple convolutional architecture operating on transition tuples to provide the most controlled comparison that we can. It is an interesting direction for future work to see how our insights in the Markovian case could be leveraged to inform sequence level models of partially observed problems.

Another line of work avoids pretraining representations directly and instead meta-learns a policy that can adapt to new tasks [Duan et al., 2017, Finn et al., 2017a,b, Yu et al., 2018, Rakelly et al., 2019, Mitchell et al., 2021]. This approach is beyond the scope of this paper which focuses on representation learning. Moreover, these meta-learning algorithms require the pretraining trajectories to be partitioned by task so that each task has multiple trajectories. Since we focus on pretraining data where we don't have access to the latent context, it is unclear how to create these meta-training datasets.

Finally, recent work has shown the promise of zero-shot generalization for multitask imitation, especially when the task identifying information is expressed in natural language to leverage advances in language models [Ding et al., 2019, Jang et al., 2022, Cui et al., 2022, Brohan et al., 2022]. This is an exciting line of work, but beyond the scope of this project where we focus on data where the context information is latent. It is an interesting direction for future work to assess precisely how much performance can be improved via extra context information to gauge whether it is worth the cost of labeling trajectories with context information.

It is an interesting direction for future work to try to better synthesize some of the findings from across this broad array of approaches to pretraining in slightly different settings.

# B  Extended experimental results

In this section we present the experimental results that were excluded from the main text due to space constraints. In particular, Section B.1 presents representation analysis by predicting one representation from another, Section B.2 presents the per-dataset results of various sweeps over dataset size and type, Section B.3 presents per-dataset results for representation analysis, and Section B.4 presents results of an ablation over multistep dynamics.

## B.1  Cross-representation prediction

In the main text, we evaluated representation quality by measuring accuracy of small MLPs to predict either the actions on the finetuning data or the low dimensional states on the pretraining

data. Here we present a similar analysis, but now where we use small MLPs to predict the other representations themselves. This is interesting since it lets us assess which representations contain enough information and shared structure to predict the other representations. Hypothetically, a representation that is easily able to recover another representation may be preferable since it retains more information.

Results presented in Figure 9 show the average across datasets of the cross-representation prediction error on a validation set from the pretraining distribution (normalized by the mean prediction error on each dataset). There are several possible takeaways from this experiments. First, looking a the rows, which correspond to the error when each method is used as the source, we can see that inverse dynamics generally has the lowest average error for predicting the other representations. This suggests that inverse dynamics is doing a good job of recovering the information that is shared among all the representations. Second, looking at the columns, which correspond to error when each representation is used as the target, we see that BC is the most difficult to predict and inverse dynamics is second most difficult. This is a somewhat surprising result, but suggests that these representations actually contain information that may have been thrown away (or at made least difficult to access via small MLP) within the other representations. Finally, note that the contrastive learner is both the worst source and easiest target, which is consistent with the idea that those representations are losing important task-relevant information.

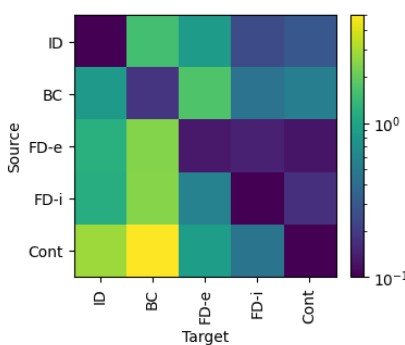

Figure 9: Cross-representation prediction error of a small MLP on a validation set from the pretraining distribution. Results are normalized per dataset by the mean error on that dataset and then averaged across datasets.

Full results on each dataset can be found in Appendix B.3 and full methodological details can be found in Appendix C.

## B.2 Per dataset evaluation success results

In the main text and Section B.1 we have only presented aggregate results that average across datasets. These averages make it easier to summarize comparisons between methods, but they sacrifice the precision of how the results vary across datasets. In this section we present per dataset results for all of the relevant sweeps across dataset variations including pretraining size, finetuning size, and finetuning size when we ablate in distribution contexts or observability of the context in the observation.

**Pretraining size.** First, we present the full ablation over pretraining size, corresponding to the right panel of Figure 3. The full per dataset results are shown in Figure 10.

There are several findings in the dataset-specific results that are not visible in the aggregate reported in the main text:

- First, the kitchen environment is a clear outlier mainly due to the stochasticity in the data generating process and smaller dataset size compared to the others (see Appendix C.1 for more detailed description of the data). As a result of the noise added to the low dimensional states, training from States actually underperforms training from Pixels + Aug. We hypothesize that this is due to some implicit regularization that arises from training from the rendered noisy observations instead of the low dimensional noisy states. Importantly, inverse dynamics is much better able to handle the stochasticity than the alternative methods given the relatively small pretraining dataset and is the only method that is able to perform comparably to training from scratch.

- Point mass is the only environment where the externally pretrained representations (R3M and Imagenet) substantially outperform training from Pixels + Aug and they are substantially outperformed on kitchen and the metaworld datasets. We hypothesize that this shows how it is quite difficult to transfer features across domains and see consistent benefits on challenging tasks.

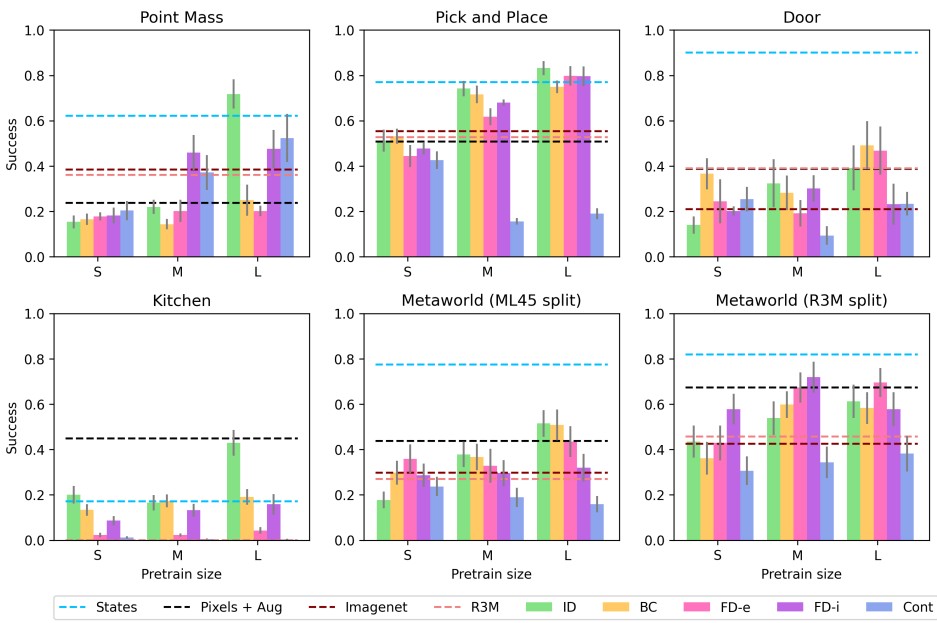

Figure 10: The per dataset results of sweeping over pretraining size, corresponding to the right panel of Figure 3. Error bars show standard error over seeds and contexts (as described in Table 1). Horizontal lines indicate mean performance of algorithms that do not depend on pretraining size.

- Note that performance of contrastive learning is substantially better relative to the alternatives on point mass. We hypothesize that this is due to the fact that random crop augmentations are actually a reasonable simulation of the dynamics in the pointmass environment specifically so that contrastive learning becomes more similar to implicit forward dynamics.

**Finetuning size.** Next, we present the full ablation over finetuning size, corresponding to the left panel of Figure 3. The full per dataset results are shown in Figure 11.

Again, as described above, Kitchen is a clear outlier due to stochasticity with inverse dynamics the best performer. Inverse dynamics is also the clear winner on point mass and a slight winner on pick and place. The other tasks are more ambiguous with many methods performing about the same, and none substantially better than training from scratch (across all pretraining sizes). Disaggregating the results here shows how even though inverse dynamics is clearly the best in aggregate, this is not necessarily true on every dataset. As we will see in Figure 12, we hypothesize that much of this weak performance can be attributed to the fact that the evaluation contexts in door and the two metaworld variants are truly out of distribution, making it difficult for any pretraining method to generalize.

**Ablating in distribution contexts.** Next, we present the full per dataset results when we ensure that all the evaluation contexts are included in the pretraining distribution, corresponding to Figure 4 in the main text. The full per dataset results are shown in Figure 12.

It is important to compare these results to those that include out of distribution evaluation contexts in Figure 11. First, note that the evaluation contexts on point mass and pick and place were already in distribution, so they are kept the same. However, on door and the two metaworld splits there is a *substantial* improvement, especially for inverse dynamics and BC. This shows how these methods can benefit from being applied on tasks that are contained in the pretraining distribution. Interestingly, even though the evaluation contexts are now in distribution, the forward dynamics representations do not see substantial improvements and are still outperformed by training from scratch on the more challenging datasets.

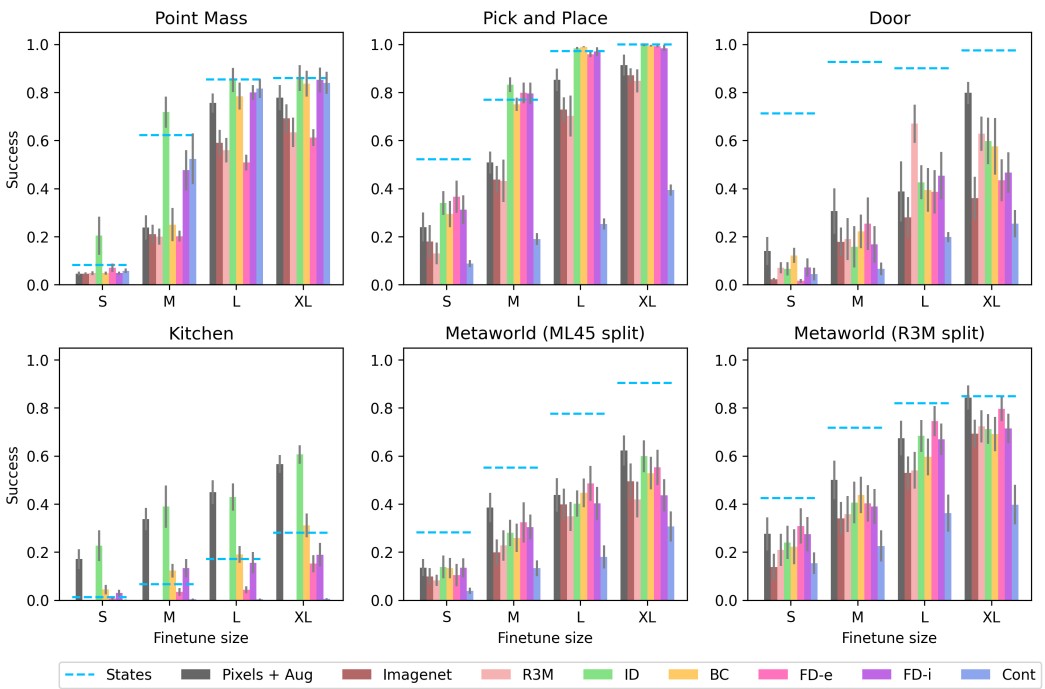

Figure 11: The per dataset results of sweeping over finetuning size, corresponding to the left panel of Figure 3. Error bars show standard error over seeds and contexts (as described in Table 1).

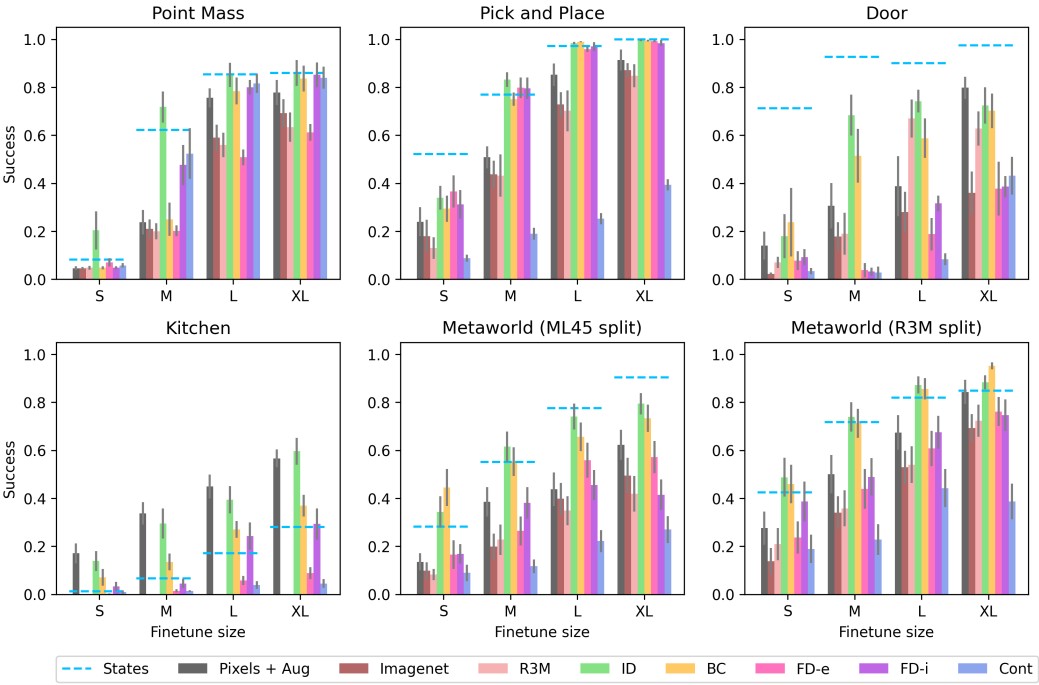

Figure 12: The per dataset results of sweeping over finetuning size when we include the evaluation tasks in the pretraining data, corresponding to Figure 4. Error bars show standard error over seeds and contexts (as described in Table 1).

**Aggregating based on context observability.** Finally, we present the full results for aggregations across whether the context is observable, corresponding to Figure 5 in the main text. Context is latent in point mass, pick and place, and kitchen, but inferrable in door and both metaworld splits. The results are shown in Figure 13. Note that these results are just grouped averages over the results presented in Figure 11.

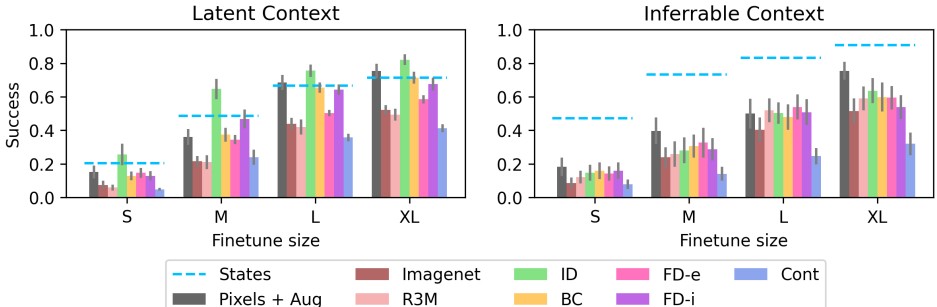

Figure 13: The full results of aggregating based on the observability of the context variable, corresponding to Figure 5. Error bars show standard error over seeds and contexts (as described in Table 1) then averaged across datasets.

Compared to Figure 5, we now include the results from all algorithms and also from the environments where the context is inferrable. As reported in the main text, there is a clear gap between inverse dynamics and BC when the context is latent, likley due to confounding. Here we see that this gap largely disappears in the datasets where the context is inferrable and generally the disparities between methods shrink.

### B.3 Per dataset representation analysis

Now we present the per dataset results of the various methods of representation analysis based on predicting different target quantities of interest: the action, the low dimensional state, and the other representations themselves.

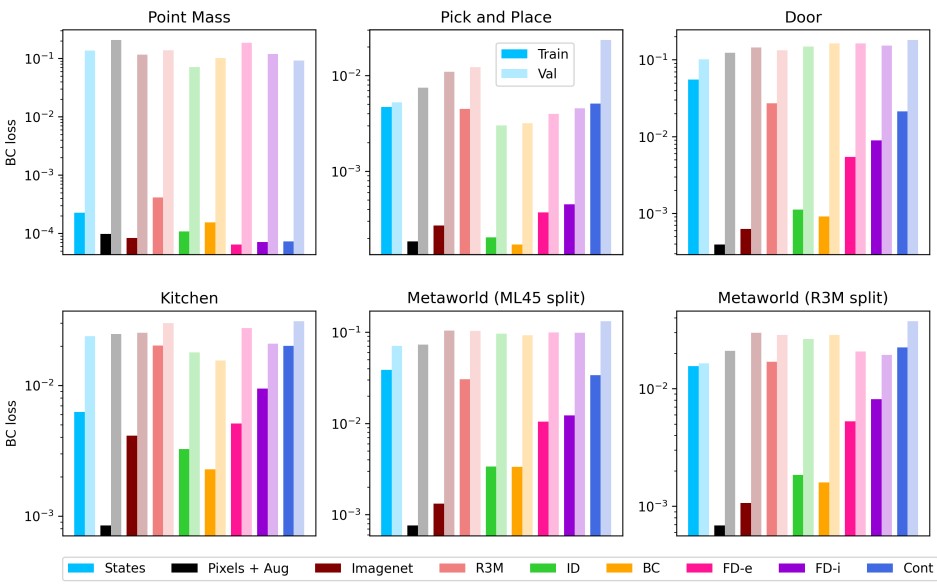

Figure 14: Full per dataset results of action prediction on the finetuning distribution.

**Predicting action.** First we present the per dataset results for train and validation action prediction on the finetuning datasets using the default pretraining and finetuning size. These results correspond

to Figure 6 from the main text. Unlike in the main text, here we do not do any normalization of the losses, so the losses occur at different scales on each dataset depending on how difficult the prediction task is. Results are shown in Figure 14.

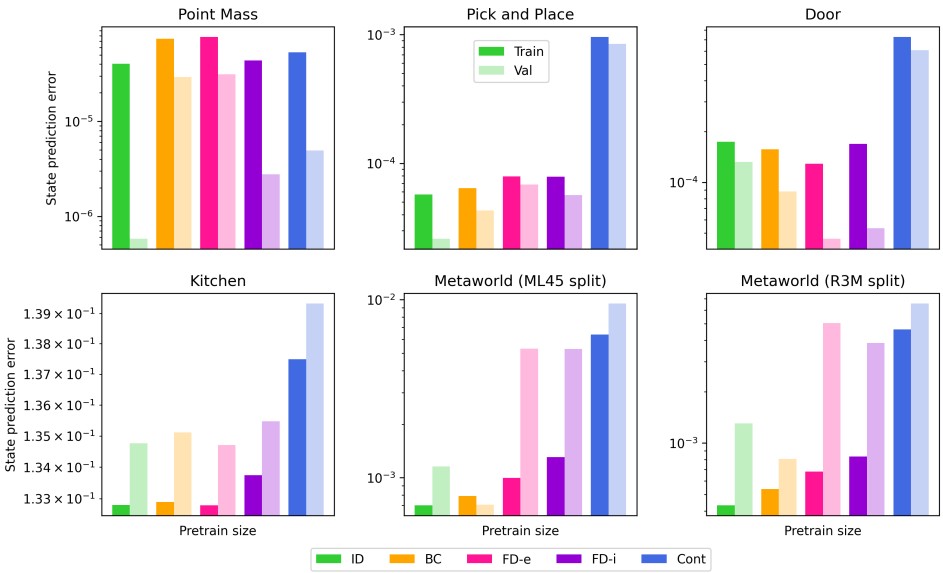

Figure 15: Full per dataset results for state prediction on the pretraining distribution.

**Predicting state.** Next, we present the per dataset results for predicting the low dimensional state on the pretraining distribution from the various learned representations. These results correspond to Figure 7 in the main text. Again, unlike in the main text, results are not normalized, so they occur at different scales across environments. Results are shown in Figure 15.

Note that as mentioned before, there is stochasticity added to the low dimensional states in the kitchen environment. This makes it difficult for any of the methods to substantially outperform the floor set by the noise level.

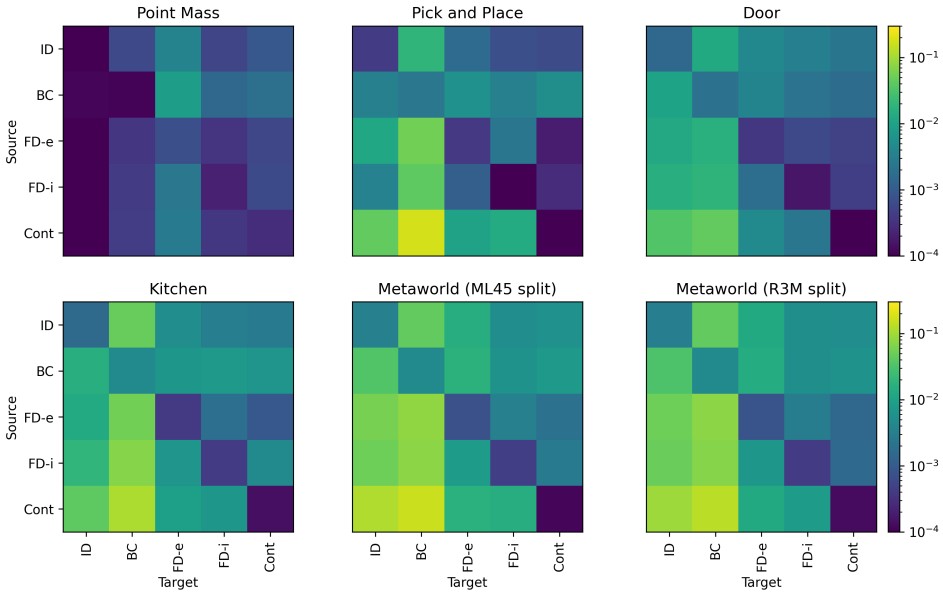

Figure 16: Per dataset results for cross-representation prediction on the pretraining distribution. Color shows the validation error of predicting target from source.

**Predicting across representations.** Finally, we present the per dataset results for predicting across the different learned representations on the pretraining distribution. These results correspond to Figure 9. Again, unlike in the averaged figure, this figure is not normalized, so the scales vary across datasets. We truncate the color scale at 1e-4 on the low end for easier visualization.

## B.4 Ablation of multistep dynamics

As mentioned in the main text, some work argues for multistep dynamics models [Efroni et al., 2021, Lamb et al., 2022]. Note that this work focuses on settings with exogenous noise which are different from the simpler settings that we consider. To confirm that using multistep dynamics models does not help to learn better representations, we run an ablation of the number of steps included in the dynamics model on three environments: point mass, pick and place, and door and two algorithms: inverse dynamics and implicit forward dynamics. Results are shown in Figure 17. At a high level, we basically find little difference when ablating the number of steps, so we default to using one step models everywhere for simplicity.

Note: for inverse dynamics models, we learn a $k$ step model by predicting $a_t$ given $o_t$ and $o_{t+k}$. For forward dynamics, we learn a $k$ step model by predicting $o_{t+k}$ given $o_t$ and $a_{t:t+k}$.

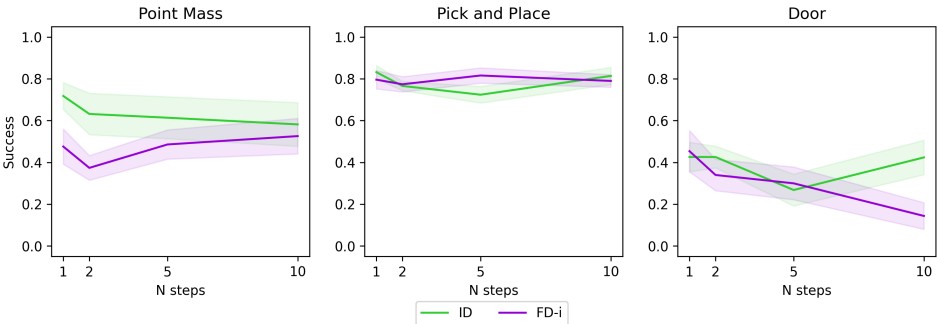

Figure 17: Sweep over the number of timesteps included in the dynamics models.

## C Detailed experimental methodology

In this section we present a detailed account of out methodology. We also release our code that was used to perform the experiments for full transparency. We split up the description into Section C.1 which describes the environments and dataset generation, Section C.2 which describes the details of the pretraining pipeline, and Section C.3 which describes the details of the finetuning and evaluation pipeline.

All code is at `https://github.com/davidbrandfonbrener/imitation_pretraining`.

### C.1 Envionment and dataset details

**Software dependencies.** All of our environments are based on the MuJoCo simulator [Todorov et al., 2012]. The point mass environment is derived from the DM control suite [Tunyasuvunakool et al., 2020]. The kitchen environment and dataset was introduced in Gupta et al. [2019]. The rest of the environments are taken from Metaworld [Yu et al., 2020]. We describe each environment in detail and summarize the descriptions in Table 2

**Point mass.** The point mass environment consists of an actuated point mass on a 2d plane. In our version, the context $c \in \mathbb{R}^2$ determines the goal location. Then, the demonstration policy $\pi_c^*$ is a PD controller that moves the point from the current position $x$ to the goal position $c$. Because the context variable is continuous, we sample an independent context for each trajectory in the pretraining dataset from the uniform distribution over possible goal states. The context is fully latent and not observable in the observation. The low dimensional state is the 2d position and the high dimensional images are 84x84x3.

**Pick and place.** The pick and place task is taken from the metaworld suite. In our version, the context $c \in \mathbb{R}^3$ determines the goal location for the block. The demonstration policy $\pi_c^*$ is a scripted policy from the metaworld repo. We remove the goal indicator from the image in this environment so that the context is fully latent and not observable from the observation. The low dimensional state is the 3d position of the gripper, 1d openness of the gripper, and 7d position and orientation of the block. The high dimensional observations are images of size 120x120x3.

**Door.** The door environment is also taken from the metaworld suite. In our version, the context $c \in [4]$ determines the index of the environment from door-close, door-open, door-unlock, and door-lock. For our default experiments we use door-close, door-open, and door-unlock as the pretraining contexts and door-lock as the eval context. For the ablation where we ensure that the eval context is in the pretraining distribution, we include door-lock in the pretraining data. The demonstration policy $\pi_c^*$ is a scripted policy from the metaworld repo. Given the context, the initial position of the robot, initial position of the door, and goal position (which is visible in the observation image) are all randomized. Note, the context is inferrable since the initial position of the door and lock allow the learner to infer the context. The low dimensional state is the 3d position of the gripper, 1d openness of the gripper, 7d position and orientation of two objects in the scene, and 3d goal position. The high dimensional observations are images of size 120x120x3.

**Kitchen.** The kitchen environment and dataset are taken from Gupta et al. [2019]. Each trajectory contains a sequence of four tasks in a simulated kitchen collected by a human demonstrator. In our version, the context $c \in [24]$ is determined by the sequence of four tasks contained within the demonstration trajectory (of which there are 24 possibilities). We evaluate on three contexts: microwave-kettle-light switch-slide cabinet, bottom burner-top burner-slide cabinet-hinge cabinet, and kettle-bottom burner-top burner-light switch. In our default setup, we pretrain on the other 21 contexts, and in the ablation of in distribution evaluation we pretrain on all 24 contexts. The context is fully latent and not observable from the initial state. The low dimensional state is a 9d description of the arm position and a 21d description of the position of objects in the kitchen. The high dimensional observations are images of size 120x120x3.

Note: the kitchen environment is the only one that we consider that has added noise. The raw data from Gupta et al. [2019] contains gaussian noise added to the low dimensional states and actions, so this noise cannot be removed without re-generating the data. We render the images from the noisy states, so there is also noise present in the image observations. We also evaluate in an environment with the same noise added, so there is no gap between training and eval.

**Metaworld (ML45 and R3M).** Finally, we consider two variants of the full metaworld suite. Here the context $c \in [50]$ determines which metaworld task is used. We consider two different train-eval splits for our default environments. The ML45 split takes the eval tasks from the original metaworld ML45 task which are bin-picking, box-close, hand-insert, door-lock, and door-unlock. The R3M split takes the eval tasks that were chosen in the R3M paper [Nair et al., 2022]: assembly, bin-picking, button-press, drawer-open, and hammer. Given the context, the initial and goal positions are randomized. The goal position is visible in the observation. The low dimensional state is the 3d position of the gripper, 1d openness of the gripper, 7d position and orientation of (potentially) two objects in the scene, and 3d goal position. The high dimensional observations are images of size 120x120x3.

Table 2: A summary of the description of datasets above. Inferrable refers to whether the context is observable. OOD refers to whether the evaluation context is out of distribution.

| Dataset | Policy | Context | Inferrable | OOD | Noise | State dim |
|---------|--------|---------|------------|-----|-------|-----------|
| Point mass | PD controller | $\mathbb{R}^2$ | No | No | No | 2 |
| Pick and place | Script | $\mathbb{R}^3$ | No | No | No | 11 |
| Door | Script | [4] | Yes | Yes | No | 21 |
| Kitchen | Human | [24] | No | Yes | Yes | 30 |
| Metaworld-ML45 | Script | [50] | Yes | Yes | No | 21 |
| Metaworld-R3M | Script | [50] | Yes | Yes | No | 21 |

## C.2 Pretraining details

**Software dependencies.** We implement all of our training in JAX [Bradbury et al., 2018]. We use flax for neural networks [Heek et al., 2023] and optax for optimization [Babuschkin et al., 2020]. Our code is loosely based on Kostrikov [2022].

**Architecture.** All of our pretraining algorithms share exactly the same encoder architecture to ensure that we have a fair comparison. Since our tasks are relatively simple visually, and so as to allow for large scale experiments without too much compute, we use a relatively small convnet encoder. Specifically, we follow the architecture from Yarats et al. [2021] which consists of a 4 layer convnet with 3x3 filters, number of channels of (32, 64, 128, 256), and strides of (2,2,1,1). We add a modification to include a spatial softmax activation [Finn et al., 2016], which we found to be important for the manipulation tasks we consider. This is followed by a linear layer to project into the embedding dimension of 64 and finally a layernorm and tanh activation to normalize the embedding. We use the gelu activation function throughout.

Now we will birefly describe the architecture used for each pretraining algorithm, following their descriptions in Section 4.2:

- Inverse dynamics: the inverse dynamics head is an MLP that takes in $\phi(o), \phi(o')$ and produces an estimated action. This MLP has two hidden layers of width 256 and dropout of 0.1 during training.

- BC: the BC policy head is an MLP with two hidden layers of width 256 and dropout of 0.1 during training.

- Implicit forward dynamics: the implicit forward dynamics model uses an action encoder $\phi_a(a)$ which outputs a 64 dimensional normalized action embedding which is concatenated to $\phi(o)$ to form $\phi(o, a)$. Then there are two projection heads $f_1, f_2$ that take in $\phi(o, a)$ and $\phi(o')$ respectively and produce 64 dimensional embeddings that are normalized to have unit norm. All these networks ($\phi_a$, $f_1$, and $f_2$) are MLPs with two hidden layers of width 256 and the relevant input and output dimensions.

- Explicit forward dynamics: the explicit forward dynamics model uses the same architecture to encode $a$ with $\phi_a$. Then, instead of projection heads, we require a convolutional decoder to produce an image. Following Yarats et al. [2021] we use an architecture that inverts the encoder, having a dense projection layer followed by channels of (256, 128, 64, 32) and strides of (1,1,2,2).

- Contrastive: the contrastive network is the same as the implicit forward dynamics network except that there is no action input and $o'$ is replaced by an augmentation of $o$.

**Training hyperparameters.** For pretraining, we split the datasets into two categories: easy (point mass, pick and place, and door) and hard (kitchen, metaworld-ml45, and meatworld-r3m). On the easy tasks we train for 100k gradient steps and on the hard tasks we train for 200k gradient steps. Batch size is 256 for all methods except explicit forward dynamics where (due to the added compute required for the decoder) we use batch size of 128 to even out computational requirements across methods. All methods are trained with the adamw optimizer with learning rate 1e-3, a cosine learning rate decay schedule, and default weight decay of 1e-4.

**Data augmentation.** Following [Chen et al., 2022] and others, we note that cropping augmentations are the most important for training policies in simulated visual domains. As such, all of our pretraining algorithms (and the Pixels + Aug baseline) use random cropping augmentations, and we found this to be an important implementation detail. The one exception is explicit forward dynamics where we found it difficult to reconstruct images with augmentations, so we omit them for that algorithm.

**Compute resources.** Pretraining was all done on an internal cluster using RTX8000 GPUs. We estimate that the final training run needed to produce the results in the paper took approximately 600 GPU hours.

## C.3 Finetuning and evaluation details

**Training hyperparameters.** The policy is always an MLP with two hidden layers of width 256. We use gelu activation and apply dropout with probability 0.1 during finetuning. We finetune on every

dataset for 10k gradient steps with batch size 256. All policies are trained with the adamw optimizer with learning rate 1e-3, a cosine learning rate decay schedule, and default weight decay of 1e-4.

As explained in Table 1 there are several seeds and evaluation contexts for each environment. For example, for the default results in Figure 1 we end up having a total of 80 different finetuning datasets per representation when sweeping across dataset, context, and seed so that Figure 1 is reporting aggregate results across 720 finetuning and evaluation runs.

**Evaluation hyperparameters.**   Each evaluation is run for 100 episodes in the environment to estimate the success of the policy (except for the kitchen environment where we run 50 episodes due to slow rendering of that environment).

**Compute resources.**   Finetuning and evaluation was all done on an internal cluster on CPU (since the finetuned policy network is small and environments run on CPU). We estimate that all the finetuning and evaluation in the final runs used to produce results for the paper took approximately 2000 CPU hours.

### C.4   Comparison to R3M experimental setup

There are several low-level but important differences between our evaluation setup and the one used in the R3M paper [Nair et al., 2022] which uses some similar environments. These differences end up making the pretrained R3M representations perform worse in our evaluations than those in the original paper. For the kitchen tasks in particular, the biggest difference is that while the R3M paper considers only learning single subtasks (e.g. slide the door open, see section 4.2 of the R3M paper), we consider learning sequences of subtasks (e.g. open the microwave, put the kettle on, turn on the light, and slide the door open, all in one trajectory). The R3M paper considers explicitly easier tasks. We did this because the kitchen data itself contains sequences of subtasks, not single subtasks (following the paper that introduced the kitchen dataset). For the metworld tasks, R3M chose to evaluate on a particular subset of tasks that are somewhat easier than average (this is why we consider two different splits of metaworld on with the R3M eval tasks and one with the original eval tasks from the metaworld paper). Another difference is that to focus solely on feature learning, we only pass in the image observation and not the proprioception while R3M passes in both. Again this makes the problem a little bit more difficult. wW also render images at a lower resolution due to computational constraints.

Finally, it is important to note that R3M is attempting to solve a different problem of general image representation learning that transfers across domains, while we are focusing on within domain, but cross-task generalization (which is easier to analyze in a controlled way).

## D   Extended analysis discussion

Here we provide a more detailed discussion of related theoretical work.

One recent line of work focuses on learning representations for exploration [Efroni et al., 2021, Lamb et al., 2022] and offline RL [Islam et al., 2022] in the presence of *exogenous noise*. The exogenous noise setting means that the high dimensional observations contain information that is not effected by the actions; e.g., background dynamics that appear in image observations but do not affect the task. This line of work argues that inverse dynamics modeling is the best approach to ignore exogenous noise. Our results are complementary to this line of work in showing that even in settings *without* exogenous noise, inverse dynamics is still often preferable to alternatives for representation learning. Moreover, we consider a *multitask* imitation setting with latent contexts while they consider single task and reward-directed problems.

Another line of work proves that learning a forward dynamics model is a well-motivated approach for multitask imitation [Nachum and Yang, 2021]. While that work does not directly compare to inverse dynamics pretraining, we find that inverse dynamics pretraining outperforms forward dynamics modeling in our settings. Moreover, while this paper shows that if our representation learns a good forward dynamics model that it works well for imitation, it does not discuss how efficiently such a representation can be learned. So, while both methods are well-motivated, we find inverse dynamics modeling to be more efficient than learning the forward dynamics.

Finally, another line of work studies multitask representation learning for imitation by directly performing behavior cloning [Arora et al., 2019, Zhang et al., 2022]. These methods provide positive results for the approach, but require algorithms that have access to the latent context information which must be discrete so as to learn a separate policy for every pretraining context, thus avoiding confounding. This method requires extra information and is difficult to scale to very large numbers of contexts. In contrast, we find that inverse dynamics modeling is able to perform well without this extra information or added complexity of learning multiple models and naturally avoids confounding by the latent context information.

