# OpenReview forum: "Inverse Dynamics Pretraining Learns Good Representations for Multitask Imitation"
_NeurIPS.cc/2023/Conference — NeurIPS 2023 poster_

### Official Review · Reviewer_Rymd · 2023-06-30

**Soundness:** 3 good
**Presentation:** 3 good
**Contribution:** 2 fair
**Rating:** 6
**Confidence:** 3

**Summary:**

This paper provides both empirical evidence and theoretical arguments for the usefulness of the inverse dynamics criteria as a pre-training objective for multitask imitation learning setups. Furthermore, the authors were able to go beyond the final conclusion and provide insightful analysis where they were able to show that the advantages of the inverse dynamics objective compared to alternative objectives such as behavioral cloning are rooted in better out-of-distribution generalization and robustness to environments with fully latent context.


**Strengths:**

1. The authors support their claims with both compelling empirical evidence and novel theoretical arguments.

2. The paper is clearly written and accessible to readers with varying backgrounds.

**Weaknesses:**

The superior performance of the inverse dynamics criteria in the examined setting is somewhat expected. Namely, unlike behavioral cloning, the latent multitask performance does not hurt it since the inverse dynamics criteria is oblivious to the latent reward. And the examined architecture seems more natural for inverse dynamics than for forward dynamics. Arguably, forward dynamics could benefit from an architecture that can fuse the observation with the action earlier.


**Questions:**

Can you elaborate on how your work relates to recent works on multitask imitation learning via generative pre-training on trajectories? It seem that the generative pre-training objective in these works implicitly combines the forward dynamics and behavioral cloning criteria.

[1] Kuang-Huei Lee, Ofir Nachum, Mengjiao Yang, Lisa Lee, Daniel Freeman, Winnie Xu, Sergio Guadarrama, Ian Fischer, Eric Jang, Henryk Michalewski, and Igor Mordatch. Multi-Game Decision Transformers, 2022.

[2] Scott Reed, Konrad Zolna, Emilio Parisotto, Sergio Gomez Colmenarejo, Alexander Novikov, Gabriel Barth-Maron, Mai Gimenez, Yury Sulsky, Jackie Kay, Jost Tobias Springenberg, Tom Eccles, Jake Bruce, Ali Razavi, Ashley Edwards, Nicolas Heess, Yutian Chen, Raia Hadsell, Oriol Vinyals, Mahyar Bordbar, and Nando de Freitas. A Generalist Agent, 2022.


**Limitations:**

In its current state, the theoretical arguments lack formal claims and proofs. Thus they feel more like mathematical intuitions rather than solid theoretical results.

---

> ### Author Rebuttal · Authors · 2023-08-08
>
>
> First, we would like to thank the reviewer for their positive comments about the compelling evidence, novel analysis, and clear writing in the paper.
>
> Here we will address each of the weaknesses and the questions raised by the reviewer. Hopefully these comments can provide some additional clarity. If they do, we encourage the reviewer to increase their score, and otherwise are happy to answer any follow up questions.
>
> 1. (Somewhat expected results) We agree with the reviewer that the results can be explained well in hindsight, and indeed this is what our analysis section intends to do. However, we disagree that these results are obvious or expected since they are not found in the existing literature (as detailed in Section 2 and Appendix A). As a side remark, we were not initially expecting to see a significant advantage of inverse dynamics ourselves at the beginning of the project. If the reviewer could point to prior work in the literature that made these observations, we would be happy to consider it, but we do not think this is a valid weakness of the work.
>
> 2. (Architectural details) We agree with the reviewer that it is difficult to make comparisons between algorithms that require different architectures. However, we disagree that our choices were in any way unreasonable. We attempted to maximize performance of all baselines with a consistent compute budget across methods. We control for architectural differences as much as possible by using the exact same image encoder architecture and feature fusion method for all networks. To compare inverse dynamics and implicit forward dynamics explicitly, both networks consist of an MLP on top of the concatenation of two encoded vectors: $ [\phi(o), \phi(o')] $ for inverse dynamics and $ [\phi(o), \phi_a(a)]$ for forward dynamics. Then the MLP head has the capacity to fuse the features. Moreover, since the image needs to be encoded with a convnet, this seems to be the standard way to incorporate action information since it cannot be fed directly into the convnet. Note for explicit forward dynamics we need to add a decoder on top of encoded features to produce an image to compute the loss. The objective requires this larger network, but we think we handled the difference in the most carefully controlled way possible. If the reviewer can point to related literature or more specific arguments about why our architectural decisions did not make sense, we would be happy to discuss further.
>
> 3. (Question - generative pretraining) Thanks for this interesting question. Indeed, it does seem that generative pretraining could be thought of as a combination of BC and FD. It is not clear why creating such a combination would resolve the issues of either method in our particular setting, but this raises an interesting point that all of the objectives we consider can be combined in various ways (e.g. added together). Anecdotally, we did try adding together various objectives during development for this project but found the mixtures to perform worse than the simple objectives. In the end, in this paper we want to focus on presenting the cleanest controlled results that we can, so we opt to exclude these combinations for now, but it is definitely an important direction for future work to consider whether cleverly combining objectives could outperform the individual ones. We will add the papers that the reviewer referenced to the related work section as well as an expanded discussion of this direction of future work.
>
> Also, as a note, we agree with the reviewer that the analysis section is not formal theory (as in theorems). It also only considers a substantially simplified model. We include it for intuition (which we believe the reviewer agrees is useful, based on the strengths section of the review). That said we will discuss how the lack of formal theorem statements is a limitation in a new limitations subsection to be included with the discussion at the end of the paper, and could present a fruitful direction for future theoretical work.

---

> > ### Comment · Reviewer_Rymd · 2023-08-20
> >
> > Thank you for the response. I do not have any major concerns. However, I have decided to keep my original score. This is because the theoretical arguments lack formal claims and proof. Additionally, the architecture for forward dynamics is quite different from that used in successful generative pre-training.

---

### Official Review · Reviewer_8bvo · 2023-07-05

**Soundness:** 3 good
**Presentation:** 3 good
**Contribution:** 3 good
**Rating:** 6
**Confidence:** 3

**Summary:**

This paper analysis the downstream fine-tuned performance on various multi-task settings after pretaining with different objectives including -

1) Inverse dynamics
2) Contrastive
3) Forward Dynamics - Explicit
4) Forward Dynamics - Implicit
5) Behaviour Cloning

They compare the performance to a model trained from scratch using image representation and augmentations (Pixels + Aug). They also compare to R3M and imagenet pretrained models as baselines. The main conclusion of this paper is that IDM works better than other pretraining objectives and also works better than models trained from scratch in small data regimes. They experimentally and theoretically show that IDM can recover the representation of the true ground truth state while other methods such as Forward dynamics and Behavior Cloning can fail to do so.


**Strengths:**

The paper presents a comprehensive evaluation of different pretraining objectives, including a comparison to models trained from scratch using image representation and augmentations (Pixels + Aug), as well as R3M and imagenet pretrained models as baselines. The main conclusion drawn from this study is that IDM (Input-Dependent Memory) outperforms other pretraining objectives, as well as models trained from scratch, especially in scenarios with limited data availability. The strengths of this paper lie in its valuable insights and experiments that not only demonstrate the superior performance of IDM but also provide a clear understanding of why it works through various experimental analyses. These findings contribute to a better understanding of pretraining objectives and their effectiveness in recovering the representation of the true ground truth state, highlighting the limitations of alternative methods such as Forward dynamics and Behavior Cloning, which fail to achieve comparable results.


**Weaknesses:**

I think the paper lacks certain details and contexts in various places. I am not sure what is the objective used while training the policy in the finetuning phase. It seems to be that it is behavior cloning but I am not sure. Another confusion I have is the inverse dynamics objective is specified using a mean-squared error loss so it seems that the actions are continuous valued but they evaluate action predictions using binary cross-entropy loss, so are action values constrained between 0 and 1. It would be useful for readers if such details are clarified.

From my understanding, the approach relies on expert trajectories to obtain pretraining data which seems like a limitation to me since such data may not be available in many cases. Many approaches like decision transformer (https://arxiv.org/abs/2106.01345) train using suboptimal data. It would be nice to have an analysis in the paper which uses suboptimal data and shows how it affects the different pretraining objectives.


**Questions:**

- It is not clear to me what context variables mean in the context of experiments. Could you clarify with examples what would be a context variable in some datasets considered in the paper?
- There is some evidence in literature in papers such as SGI and SPR that contrastive learning is useful for RL. Although their finetuning setup is different from this paper as they do not use imitation learning for finetuning. Do the authors have any intuition as to why contrastive learning helps there but not here?

SGI - https://arxiv.org/abs/2106.04799
SPR - https://arxiv.org/abs/2007.05929

**Limitations:**

Yes, the authors have addressed limitations.

If the authors use expert data for training as I pointed above, then I think they should mention that as a limitation as well as I think it might be difficult to obtain such data in many environments.

---

> ### Author Rebuttal · Authors · 2023-08-08
>
>
> First, we would like to thank the reviewer for their comments about the strengths of the paper, namely the "comprehensive evaluation" and resulting "valuable insights".
>
> Here we will address each of the weaknesses and questions raised by the reviewer. Hopefully these comments can provide some additional clarity. If they do, we encourage the reviewer to increase their score, and otherwise are happy to answer any follow up questions.
>
> 1. (Setup details) Thanks for raising these issues to our attention. Indeed, we use behavior cloning during the finetuning phase, we will clarify this more explicitly in Equation (5). For inverse dynamics, indeed we also use mean squared error. We are not sure where the reviewer sees binary cross-entropy loss in the paper since it is never used or mentioned. Actions are always continuous and always trained using mean squared error. Perhaps the confusion arises since we report task success as a percentage (valued between 0 and 1), but this is really the average reward not a measure of action prediction accuracy (where the reward is 1 for success and 0 for failure). We will clarify this in the paper.
>
> 2. (Suboptimal data) We agree that extending our results to cases with suboptimal data would be an interesting direction for future work and explicitly say so briefly in the discussion in lines 361-362. However, algorithms like decision transformer that rely on access to task rewards are beyond the scope of our paper where we instead focus on the imitation setting where there is no reward information available. We made this choice so that we could create carefully controlled experiments to just test the different representation learning algorithms without the additional complexities of reward-based learning. We will add a broader discussion of this decision in the discussion section as a great direction for future work.
>
> 3. (Question - context variables) Thanks for this question, the context variables are detailed in Appendix C, but we realize that we likely did not provide enough clarity in the main text. For example, in the pointmass task the context determines the 2d continuous goal location, in the kitchen task the context determines which of 24 possible sequences of subtasks is the desired behavior (e.g. open the microwave then put the kettle on then turn on the light then open the cabinet is an example of one specific context), and in the metaworld tasks the context determines which of 50 behaviors is desired (e.g. close the box is one context and lock the door is a different context). We will move some of these examples from the appendix into the main text to increase clarity.
>
> 4. (Question - contrastive learning) Thanks for bringing these related works to our attention. There are substantial differences between our setup and the one in SGI or SPR beyond just imitation vs. RL (although that is indeed a major difference) that could lead to different results for contrastive learning. SPR is a single-task online method which is very different in nature from our offline multitask setup that explicitly considers the ability of representations to transfer to novel contexts (i.e. tasks). SGI is more similar since they also consider offline pretraining, however instead of considering a setting where information is transferred across tasks from multitask expert pretraining data, they consider single task learning from non-expert policies by collecting data from suboptimal (or random) policies that are attempting to solve the same task. We hypothesize that this substantial difference in dataset composition and gap between pretraining and finetuning tasks is likely the reason that they find contrastive learning to be more useful. It is interesting to note that even in this substantially different setting, the SGI paper finds inverse dynamics modeling alone to be more useful that contrastive learning alone in an ablation (Table 3), so maybe our analysis could be extended to the setting they consider in SGI in future work. We can add a discussion of these paper to the extended related work in Appendix A.
>
> The biggest difference is that those papers do single task learning with auxiliary objectives, while we do multitask learning. Explicitly, SGI and SPR consider an online RL problem where there is only one task (i.e. one context) which determines the reward function. They do not attempt to re-use representations across tasks (i.e. for different reward functions). On the other hand, we focus on a multitask imitation setup where we are given a fixed dataset of offline experience from many different contexts (i.e. tasks) and learn one representation .
>
> Also, as a note, we will explicitly add some of the above discussion about suboptimal data to an explicit limitations subsection included with the discussion at the end of the paper (currently limitations are only discussed in passing, not in one central place as the reviewer noted).

---

### Official Review · Reviewer_t1mu · 2023-07-06

**Soundness:** 3 good
**Presentation:** 3 good
**Contribution:** 3 good
**Rating:** 6
**Confidence:** 3

**Summary:**

The paper studies how to effectively pretraining representations in imitation learning where we have acess to a pretraining dataset
with multi-task demonstrations with an unobserved latent context variable for each task and limited amount of finetuning demonstrations
for transferring to novel context. The goal in this setup is to learn good low dimensional representations of high dimensional input
(e.g. visual) to enable transfer to nove context for finetuning.

The paper claims, inverse dynamics modeling is a well-suited objective for imitation learning pretraining setting and provide
 empirical evidence supportng the claim. In addition, the paper also derives a theoretical analysis using simple
general environmental model.

**Strengths:**

1. The paper presents an extensive set of experiments for comparing different methods of pretraining policy representations using
multiple training methods like Inverse Dynamics, Behavior Cloning, Forward dynamics (implicit and explicit), training policy from scratch
, using pretrained visual representations and contrastive pretraining.

2. The results are presented on a diverse suite of tasks which include simpler environments like PointMass to slightly more difficult
environments like MetaWorld or Kitchen environments.

3.  The paper evaluates, how well these representations perform with respect to dataset size used for pretraining and finetuning,
 how well policies learnt from these mthods generalize to in and out of distribution evaluation tasks. Authors find that ID pretraining leads
to better performing representations on average across benchmark. The paper also presents a nice analysis on how well the representations
learned can predict state and actions. Interestingly ID performs quite well at predict all of this info vs any other method.

4. Finally, the paper also presents a theoretical analysis to support the empirical results using a simplified model


5. The paper doesn't present any novel method but presents interesting and valuable analysis that is worth sharing with the community

**Weaknesses:**

1. It is not clear why pretrained robotic representations like R3M performs so poorly on almost all of the tasks. The R3M paper
also has results on MetaWorld, Franka Kitchen tasks and it seems like the results for kitchen environment with R3M are quite
poor based on figures 11 & 12 from appendix. Can authors please clarify if the experiments use the same setup as R3M paper?
Also, it might good to provide further insights into why this is the case.

2. The evaluation of these methods is shown on different simple simulated environments but it might also be interesting to compare
these results on more photorealistic simulation environments like Habitat or AI2Thor for tasks like manipulation.

3. The performance of training from scratch baseline seems to almost always be equal to performance of policies trained using ID
pretrained representations as we scale the training dataset size. This seems a bit concerning as pretraining also requires large
amounts of data for any of these methods. Do authors have a specific set of tasks where they observe sample efficiency? i.e.
ID pretrained representations converge to similar performance early on during finetuning? It'd be good to outline the gains achieved
apart from task performance to quantify importance of pretrained representations.


**Questions:**

The paper doesn't present any novel method but presents interesting and valuable analysis that is worth sharing with the community.

As outlined in weakness section if authors can discuss about the mentioned questions.

**Limitations:**

Scaling these methods to much more complex high dimensional and long-horizon tasks might not be trivial given the compute
requirement.

No novelty in contributing a new method but the paper presents interesting analysis

---

> ### Author Rebuttal · Authors · 2023-08-08
>
> First, we would like to thank the reviewer for their thoughtful and detailed comments about the strengths of the paper on both the experimental and analytical/theoretical fronts.
>
> Here we will address each of the weaknesses raised by the reviewer in turn. Hopefully these comments can provide some additional clarity. If they do, we encourage the reviewer to increase their score, and otherwise are happy to answer any follow up questions.
>
> 1. (R3M results) Thanks for raising this subtle issue. Indeed, there are several low-level but important differences between our evaluation setup and the one used in the R3M paper. For the kitchen tasks in particular, the biggest difference is that while the R3M paper considers only learning single subtasks (e.g. slide the door open, see section 4.2 of the R3M paper), we consider learning *sequences of subtasks* (e.g. open the microwave, put the kettle on, turn on the light, *and* slide the door open, all in one trajectory). The R3M paper considers explicitly easier tasks. We did this because the kitchen data itself contains sequences of subtasks, not single subtasks (following the paper that introduced the kitchen dataset). For the metworld tasks, there is a similar pattern where R3M chose to evaluate on particularly easy tasks (this is why we consider two different splits of metaworld on with the R3M eval tasks and one with the original eval tasks from the metaworld paper). Another difference is that to focus solely on feature learning, we only pass in the image observation and not the proprioception while R3M passes in both. Again this makes the problem a little bit more difficult. Finally, we also render images at a lower resolution due to computational constraints. When applying R3M we resize the images, but they will be more pixelated than the ones R3M was originally evaluated on. All of these differences likely contribute to the results that we report. We will add a discussion of these differences to the appendix in the paper. As one last point, it is important to note that R3M is attempting to solve a different problem of general image representation learning that transfers across domains, while we are focusing on within domain, but cross-task generalization (which is easier to analyze in a controlled way).
>
> 2. (Photorealistic environments) We agree that evaluation in more challenging environments would be an interesting extension of our work (as we briefly state in the discussion in lines 360-361). However, we would argue that the tasks that we consider are not toy, and do provide sufficient complexity for interesting results. The main goal of this paper was to create carefully controlled and targeted experiments in simple domains to get clear insight, and we think we have accomplished this. Moreover, we prioritized domains with large numbers of predefined tasks and datasets with a single morphology that had been used in related work. We are aware of Habitat and AI2-Thor, but have not seen similar suites of tasks with demonstration data as exists for the environments that we chose (and as a result, none of the related work uses these environments either). That said, we definitely agree that future work to scale up these insights by building better environments and datasets and trying things on real robots is indeed a good idea. We will add a broader discussion of this issue to the paper.
>
> 3. (Data scaling) As the reviewer notes, once we approach the limit of large finetuning data, the gap between pretraining and training from scratch disappears. We want to emphasize that this is totally expected behavior for testing transfer learning with any pretraining algorithm. As long as the training from scratch algorithm is sound, it will approach optimal performance if given enough data. The interesting case for pretraining (and the one that we focus on in our main experiments, e.g. in Figure 1), is when we only have small amounts of finetuning data. Figure 1 shows that in this regime, there is indeed a benefit to pretraining. But as the reviewer notes, we should indeed be careful to point out that these gains explicitly depend on the relative amounts of pretraining and finetuning data that are available as shown explicitly by sweeping these parameters in Figure 3. We will make this more clear in the paper.
>
> Also, as a note, we will explicitly add some of the above discussion about more challenging domains to an explicit limitations subsection included with the discussion at the end of the paper (currently limitations are only discussed in passing, not in one centralized place).

---

> > ### Comment · Reviewer_t1mu · 2023-08-17
> >
> > Thanks for responding to my questions. I think my concerns have been addressed in general. I'll keep my score unchanged to reflect this.

---

### Official Review · Reviewer_JZDX · 2023-07-07

**Soundness:** 3 good
**Presentation:** 3 good
**Contribution:** 3 good
**Rating:** 6
**Confidence:** 4

**Summary:**

This paper investigates the effectiveness of several popular representation learning techniques in the context of imitation learning. While there is no novel components in this paper, the main strength of this paper is that it does well in designing the experimental setups to investigate the effect of several components that might disrupt the message of experiments. And in conclusion, the paper shows that inverse dynamics modelling objective is the best objective among the ones considered in the paper.

**Strengths:**

- Crystal-clear writing that clearly explains the setup and the results in a well-structured manner.
- Extensive experimental results that show inverse dynamics modelling can be better than other objectives in a controlled way.


**Weaknesses:**

- While the results are quite conclusive in the considered setup of imitation learning with clean expert demonstrations, it might not hold in a setup where the dataset is suboptimal so that it's difficult to learn the intention of the agent for making $a_{t}$ even with the access to consecutive two observations $o_{t}$ and $o_{t+1}$. Though I understand that this is not the main goal of this paper, having more discussion on this front, along with the discussion on the paper that discusses the sufficiency of representation learning for control [1], can be useful and strengthen the paper.
- Current trend might not be conclusive as the considered domains are still very simple. Including some additional experiments on more challenging domains like RLBench [2] and RoboSuite [3], which are considered more real-world-ish than simple Kitchen or Meta-World domains, or even on real-world robotics domains (I understand this might not be available in most cases) can make the conclusion of this paper be much more convincing.
- Investigation into the effect of data configuration can be more helpful for understanding when the inverse dynamics modelling can be effective. For instance, how does the trend change when the pre-training and fine-tuning domains are more simpler domains?
- This is very minor, but including the resolution of the result figures and making them be vectorized figures (by exporting them as pdfs) in the draft can be helpful for the clarity of the paper. Also there's a typo in line 278 (the the).

[1] Rakelly, Kate, et al. "Which Mutual-Information Representation Learning Objectives are Sufficient for Control?." Advances in Neural Information Processing Systems 34 (2021): 26345-26357.

[2] James, Stephen, et al. "Rlbench: The robot learning benchmark & learning environment." IEEE Robotics and Automation Letters 5.2 (2020): 3019-3026.

[3] Zhu, Yuke, et al. "robosuite: A modular simulation framework and benchmark for robot learning." arXiv preprint arXiv:2009.12293 (2020).

**Questions:**

- Please address my concerns & questions in Weaknesses.
- Having a recurrent architecture for BC or stacking some frames might resolve the issue of BC for inferring the latent variable. Did you consider these baselines?

**Limitations:**

I couldn't find the discussion on the limitation of this paper in the submitted main draft.

---

> ### Author Rebuttal · Authors · 2023-08-08
>
> First, we'd like to thank the reviewer for their positive comments about the clarity of the paper and comprehensiveness of our controlled experiments.
>
> Here we will address each of the weaknesses and one additional question raised by the reviewer in turn. Hopefully these comments can provide some additional clarity. If they do, we encourage the reviewer to increase their score, and otherwise are happy to answer any follow up questions.
>
> 1. (Discussion of suboptimal data) We agree with the reviewer that the results do not necessarily extend to cases of suboptimal data (as we state briefly in the discussion in lines 361-362). The main reason for this choice is to keep everything self-contained in an imitation framework. Once data is suboptimal, then it makes more sense to look towards methods like offline RL which were beyond the scope of our study. We would hope that the insights from the imitation case would transfer to RL settings, but this is left to future work. We will add a broader discussion of this issue to the paper.
>
> 2. (More challenging domains) Again we agree with the reviewer that using more challenging or real-world domains would be a nice extension of the paper (as we briefly state in the discussion in lines 360-361). However, we would argue that the tasks that we consider are not toy and do provide sufficient complexity for interesting results. The main goal of this paper was to create carefully controlled and targeted experiments in diverse domains to get clear insight, and we think we have accomplished this. Moreover, we prioritized domains with large numbers of predefined tasks and datasets with a single morphology. We decided that robosuite did not have sufficiently many tasks and chose metaworld over RLBench because it had been used more frequently in related work (like R3M) and depends on mujoco rather than coppeliasim which was not supported on our compute infrastructure. That said, we definitely agree that future work to scale up these insights by building better environments and datasets and trying things on real robots is indeed a good idea. We will add a broader discussion of this issue to the paper.
>
> 3. We are not sure what the reviewer meant by "how does the trend change when the pre-training and fine-tuning domains are more simpler domains?" Any further clarification would be much appreciated.
>
> 4. (Typo and resolution) Thanks for pointing out these issues, we will fix them in the paper.
>
> 5. (Question -- frame stacking) Thanks for raising this interesting question. We did try adding frame stacking at one phase of experimentation and found that it did not help, so we did not scale it up to the full experiments. If the reviewer thinks this is a crucial issue, we can attempt to go back and run these experiments across the full suite. From the theoretical side, we do not think that adding recurrence or frame stacking will resolve the dependence issue in general. Looking at the graphical model in Figure 1(a), note that even if we condition on $ o_1 $ and $ o_2$, we do not break the connection between $ c $ and $ a_2$. However, it is possible that in certain environments the history could be sufficient to uniquely determine $ c $ and thus break the dependence. That said, just because $ c $ could be inferred does not mean that we will learn the desired features which are importantly independent of $ c$. It seems that we may instead learn features that depend explicitly on $ c $ and thus would struggle to generalize to new contexts.
>
> Also, as a note, we will explicitly add some of the above discussion about suboptimal data and more challenging domains to an explicit limitations subsection included with the discussion at the end of the paper (currently limitations are only discussed in passing, not in one central place as the reviewer noted).

---

> > ### Comment · Reviewer_JZDX · 2023-08-14
> >
> > Thank you for the response. What I meant for simpler domains were more easier ones as in [Chen et al., 2022], but I think it's not crucial. Similarly, I don't think frame stacking experiments are crucial. I have no major concerns and decided to maintain the score, considering that investigation on more challenging domains and suboptimal data are left for a future work.
> >
> > [Chen et al., 2022] Chen, Xin, Sam Toyer, Cody Wild, Scott Emmons, Ian Fischer, Kuang-Huei Lee, Neel Alex et al. "An empirical investigation of representation learning for imitation." arXiv preprint arXiv:2205.07886 (2022).

---

### Decision · Program_Chairs · 2023-09-21

**Decision:**

Accept (poster)

**Comment:**

Inverse dynamics is the superior objective function for multitask imitation. The demonstration of this and the arguments that support it will be of wide interest and will influence other work. Reviewers were unanimous in their support. At the same time, they noted that the manuscript does not provide a proof as to why this is. This could also be interesting future work.

Multiple reviewers brought up the fact that the experiments are all in simple environments. Authors responded that they believe their experiments are sufficient. There is some question about how true this is and it would be good to highlight this explicitly in the manuscript. It may be the future work finds that the picture is not as clear in more complex domains or in some domains with specific properties.